# Immune Modulation of Platelet-Derived Mitochondria on Memory CD4^+^ T Cells in Humans

**DOI:** 10.3390/ijms21176295

**Published:** 2020-08-31

**Authors:** Haibo Yu, Wei Hu, Xiang Song, Yong Zhao

**Affiliations:** Center for Discovery and Innovation, Hackensack Meridian Health, Nutley, NJ 07110, USA; Haibo.Yu@HMH-CDI.org (H.Y.); whu2@stevens.edu (W.H.); Xiang.Song@HMH-CDI.org (X.S.)

**Keywords:** platelets, mitochondria, CD4^+^ T cells, immune modulation, central memory T cells, effector memory T cells, type 1 diabetes, autoimmune disease

## Abstract

CD4^+^ T cells are one of the key immune cells contributing to the immunopathogenesis of type 1 diabetes (T1D). Previous studies have reported that platelet-derived mitochondria suppress the proliferation of peripheral blood mononuclear cells (PBMC). To further characterize the immune modulation of platelet-derived mitochondria, the purified CD4^+^ T cells were treated, respectively, with platelet-derived mitochondria. The data demonstrated that MitoTracker Deep Red-labeled platelet-derived mitochondria could directly target CD4^+^ T cells through C-X-C motif chemokine receptor 4 (CXCR4) and its ligand stromal cell-derived factor-1 (SDF-1), regulating the anti-CD3/CD28 bead-activated CD4^+^ T cells. The result was an up-regulation of Naïve and central memory (T_CM_) CD4^+^ T cells, the down-regulation of effector memory (T_EM_) CD4^+^ T cells, and modulations of cytokine productions and gene expressions. Thus, platelet-derived mitochondria have a translational potential as novel immune modulators to treat T1D and other autoimmune diseases.

## 1. Introduction

Type 1 diabetes (T1D) is one of the most common autoimmune diseases which is characterized by the destruction of insulin-producing β cells by autoreactive T cells. Patients require life-long management with daily glucose monitoring and multiple insulin injections. Clinical evidence and animal studies have demonstrated that CD4^+^ T cells play a key role in the initiation and development of T1D [1]. Specifically, autoimmune memory CD4^+^ T cells become “stumbling blocks” that hinder most attempts to treat or heal T1D and other autoimmune diseases [2,3,4]. Based on the different expressions of surface markers, such as the isoforms of leukocyte common antigens CD45RA or CD45RO and CC chemokine receptor type 7 (CCR7), human CD4^+^ T cells were categorized with CD45RA^+^CD45RO^−^CCR7^+^ naïve T cells, CD45RA^−^CD45RO^+^CCR7^+^ central memory T cells (T_CM_), and CD45RA^−^CD45RO^+^CCR7^−^ effector memory T cells (T_EM_) [5]. Due to the lack of CCR7 expression, CD4 T_EM_ cannot return to lymph/blood circulation and become the resident memory cells remaining in tissues [3,4,6,7]. These resident autoimmune memory CD4 T_EM_ cells may replicate quickly and evoke the immune system to destroy the newly-generated islet β cells. Therefore, these autoimmune memory T cells need to be eradicated in order to fundamentally correct the autoimmunity of T1D. Additionally, it will be essential to modulate CD4^+^ T cells for the treatment of other autoimmune disease such as multiple sclerosis (MS) [8] and infection diseases [9], due to their dominant roles in orchestrate the innate and adaptive immune responses.

Recent clinical trials [10,11] have highlighted the limits of conventional immune therapy and underscored the need for novel approaches that not only overcome multiple immune dysfunctions, but also help to restore pancreatic islet β cells. To address these two key issues, we have developed a unique and novel process designated the Stem Cell Educator (SCE) therapy. With this SCE technology, a patient’s blood is circulated through a blood cell separator where the patient’s lymphocytes are co-cultured with adherent cord-blood-derived multipotent stem cells (CB-SC) in vitro. The “educated” lymphocytes are then returned to the patient’s circulation [12]. Over the last 10 years, our unique technology has been evaluated through international multi-center clinical studies [12,13,14,15,16] in the United States (NCT02624804), China (NCT01350219, NCT01415726, and NCT01673789), and Spain (NCT01350219), where we have demonstrated the clinical efficacy and safety of SCE Therapy. Notably, SCE therapy is the leading approach to date to safely and efficiently correct autoimmunity and restore β-cell function in T1D patients. Previous clinical trials have demonstrated that SCE therapy provided lasting reversal of autoimmune memory T cells evidenced by the up-regulation of the percentages of Naïve T cells and CD4^+^ T_CM_ and the down-regulation of CD4^+^ T_EM_ in Caucasian T1D subjects [14]. During mechanistic studies, we found that platelets’ numbers of T1D patients were increased after receiving SCE therapy. Additionally, platelet-derived mitochondria displayed the immune tolerance-associated markers such as programmed death ligand-1 (PD-L1, CD274) and Herpes Virus Entry Mediator (CD270), stimulating the proliferation of human islet β cells [16]. The effects of platelet-derived mitochondria on CD4^+^ T cells remain unclear for the treatment of T1D.

Recently, Dache et al. and our lab discovered the existence of circulating mitochondria in human and animal blood [17,18]. This finding indicates that mitochondria act not only as organelles of “a cellular power plant” for ATP production, but also function as novel mediators contributing to the crosstalk among different cells, tissues, and organs, and homeostatic maintenance [17]. Since red blood cells (RBC) and platelets comprise 99% of the cellular component of human peripheral blood [19] and there are no mitochondria in mature RBC [20], it is likely that these circulating mitochondria are primarily released by platelets [16]. To further explore the immune modulation of platelet-derived mitochondria, the purified CD4^+^ T cells were treated with platelet-derived mitochondria, respectively. The data established significant changes in CD4^+^ T cells after the treatment with platelet-derived mitochondria.

## 2. Results

### 2.1. Suppression of Peripheral Blood Mononuclear Cells (PBMC) Proliferation by Platelet-Derived Mitochondria

Initially, the purity and quality of platelet-derived mitochondria were examined by flow cytometry with different organelle-specific markers that included MitoTracker Deep Red staining, anti-cytochrome C and anti-heat shock protein (HSP) 60 Abs for mitochondria, calnexin for endoplasmic reticulum (ER), and GM130 for Golgi apparatus. Flow cytometry demonstrated that the purity of the isolated mitochondria was ≥90%, with about 4% cytochrome C^+^GM130^+^ (Figure 1A) and 5% cytochrome C^+^calnexin^+^ cells (Figure 1B). Additionally, Western blot was performed by using the mitochondrial marker cytochrome C, which locates in the inner mitochondrial membrane and transfers electrons from complex III to complex IV to facilitate the energy production. The data demonstrated the expression of cytochrome C in the purified platelet-derived mitochondria (*n* = 8) (Figure 1C), but they were negative for ER marker calnexin. However, PBMC control displayed high level of calnexin, with low levels of cytochrome C (Figure 1C, right lane). It suggested that there was the high purity of platelet-derived mitochondria.

To explore the immune modulation of platelet-derived mitochondria, the anti-CD3/CD28 bead-activated PBMC were initially treated with different dosages of platelet-derived mitochondria ranged from 0 to 200 µg/mL. The percentage of apoptotic cells was dramatically increased at the dosage of 200 µg/mL in the mitochondrial treatment group (Figure 2A, *p* = 0.003). Next, the effects of mitochondrial treatment on anti-CD3/CD28-activated PBMC proliferation were assessed by carboxyfluorescein succinimidyl ester (CFSE) staining and flow cytometry analysis. The data demonstrated that the proliferation of anti-CD3/CD28-activated PBMC was markedly reduced from 81.2% ± 4.1% to 65.6% ± 5.3% after the treatment, with platelet-derived mitochondria at 100 µg/mL (*p* = 0.0003) (Figure 2B). In comparison, treatment with other dosages of mitochondria, such as 25 µg/mL and 50 µg/mL, failed to show the suppression of anti-CD3/CD28-activated PBMC proliferation (Figure 2B, *p* = 0.74 and *p* = 0.53, respectively).

To determine the interaction of platelet-derived mitochondria with different types of immune cells, PBMC were treated with MitoTracker Deep Red-labeled mitochondria. Different types of immune cells were analyzed after being gated with different cell lineage-specific markers such as CD3 for T cells, CD4 for CD3^+^CD4^+^ T cells, CD8 for CD3^+^CD8^+^ T cells, CD11c for myeloid dendritic cells (DC), CD14 for monocytes, CD19 for B cells, and CD56 for NK cells (Figure 2C). After an incubation for 24 h, flow cytometry demonstrated that different subsets of immune cells exhibited intensity of MitoTracker Deep Red at different levels of fluorescence (Figure 2D,E). Notably, CD14^+^ monocytes exhibited higher median fluorescence intensity (83.1 ± 10.4) of MitoTracker Deep Red-labeled mitochondria than those of other immune cells. For example, they (CD14^+^ monocytes) exhibited about two times higher intensity than that of CD11c^+^ DC (43.5 ± 1.9) (Figure 2E). Additionally, the median fluorescence intensity of CD4^+^ T cells was higher than that of CD8^+^ T cells, suggesting that platelet-derived mitochondria mainly interact with monocytes, DC, CD4^+^ T cells, and CD19^+^ B cells. Based on our previous clinical data regarding the major role of CD4^+^ T cells in T1D [12,14], the purified CD4^+^ T cells were focused and treated by platelet-derived mitochondria to further explore the molecular mechanisms underlying SCE therapy for the treatment of T1D.

### 2.2. Platelet-Derived Mitochondria Directly Interact with CD4^+^ T Cells

To determine the direct interaction of platelet-derived mitochondria with CD4^+^ T cells, the purified CD4^+^ T cells were treated with different dosages of MitoTracker Deep Red-labeled mitochondria ranging from 0 to 100 µg/mL. Flow cytometry demonstrated that the median fluorescence intensities of CD4^+^ T cells were increased in a dose-dependent manner (Figure 3A), with a significant increase at 100 µg/mL relative to 50 µg/mL (*p* = 0.00086). Confocal microscopy showed that MitoTracker Deep Red-labeled mitochondria adhered to the CD4^+^ T cells (Figure 3B), with some distributions of MitoTracker Deep Red-positive platelet-derived mitochondria in CD4^+^ T cells after the co-incubation for 2 h at room temperature. Using human mitochondrial DNA (mtDNA)-specific gene markers such as nicotinamide adenine dinucleotide (NADH), dehydrogenase 1 (ND1) and ND5, real-time PCR further confirmed that copies of mtDNA in CD4^+^ T cells were also markedly increased at the dosage of treatment with 100 µg/mL relative to the untreated CD4^+^ T cells (Figure 3C, *p* = 0.039).

Our previous work reported the expression of stromal cell-derived factor 1 (SDF-1) on platelet-derived mitochondria [21]. Human CD3^+^CD4^+^ T cells strongly express CXCR4 (Figure 3D, 99.57% ± 0.16%, *n* = 3). To define SDF-1 and its ligand CXCR4 contributing to the interaction of platelet-derived mitochondria with CD4^+^ T cells, CXCR4 antagonist AMD3100 was utilized to block the interaction between CXCR4 and SDF1 (Figure 3E). Flow cytometry results established that the percentage of MitoTracker Deep Red-positive CD4^+^ T cells was markedly decreased after the treatment with 30 µM AMD3100 for 24 h. To further confirm the action of SDF-1/CXCR4 during the immune modulation of platelet-derived mitochondria on CD4^+^ T cells, we performed blocking experiments by using the blocking Ab to SDF-1. After co-culture of CD4^+^ T cells with MitoTracker Deep Red-labeled mitochondria for 2 h, flow cytometry demonstrated that both the percentage of positive cells and their median fluorescence intensity (MFI) were all declined in the presence of the treatment with SDF-1 Ab (Figure 3F,G). The median fluorescence intensity was reduced from 44.8 ± 7.2 to 21.1 ± 2.8 after the treatment with SDF-1 Ab (*p* = 0.006). Thus, the data indicated the direct interaction of platelet-derived mitochondria with CD4^+^ T cells through SDF-1/CXCR4.

### 2.3. Effects of Platelet-Derived Mitochondria on the Differentiation of Memory CD4^+^ T Cells

To determine the modulation of platelet-derived mitochondria on memory CD4^+^ T cells, the anti-CD3/CD28-activated CD4^+^ T cells were treated with platelet-derived mitochondria for 48 h. Mitochondrion-untreated anti-CD3/CD28-activated CD4^+^ T cells served as control group. Flow cytometry revealed that the percentages of naïve CD4^+^ T cells (CD4^+^CD45RO^−^CCR7^+^) (Figure 4A,B) and T_CM_ cells (CD4^+^CD45RO^+^CCR7^+^) were markedly increased (Figure 4A,C), but the percentage of T_EM_ cells (CD4^+^CD45RO^+^CCR7^−^) was significantly decreased (Figure 4D) post treatment with platelet-derived mitochondria.

Additionally, a LEGENDplex™ Human Th Panel kit was used to examine the levels of CD4^+^ T-associated cytokines (e.g., IL-2, IL-4, IL-5, IL-6, IL-9, IL-10, IL-13, IL-17F, IL-21, IL-22, IFN-γ, and TNF-α) in the supernatants of anti-CD3/CD28-activated CD4^+^ T cells in the presence or absence of mitochondrial treatment. Flow Cytometry analysis demonstrated that the levels of IL-6 (Figure 5A, *p* = 0.0003), IL-5 (Figure 5B, *p* = 0.034), and IL-21 (Figure 5C, *p* = 0.047) significantly decreased after the treatment with platelet-derived mitochondria while the level of IL-4 (Figure 5D, *p* = 0.003) increased post the mitochondrial treatment. There were no marked differences in the levels of other cytokines such as IL-2, IL-9, IL-10, IL-13, IL-17F, IL-22, IFN-γ, and TNFα (Figure 5E–L). These data suggest that platelet-derived mitochondria display multiple immune modulations on CD4^+^ T cells.

To further substantiate the action of SDF-1/CXCR4 during the immune modulation of platelet-derived mitochondria on CD4^+^ T cells, we performed blocking experiments on the anti-CD3/CD28-activated CD4^+^ T cells by using the blocking Ab to SDF-1. Flow cytometry established that the up-regulated percentages of naïve CD4^+^ T cells (CD4^+^CD45RO^−^CCR7^+^) and T_CM_ cells (CD4^+^CD45RO^+^CCR7^+^) were markedly decreased after the treatment with platelet-derived mitochondria + SDF-1 Ab relative to those in the platelet-derived mitochondrion group (Figure 6A,B); the down-regulated percentage of T_EM_ cells (CD4^+^CD45RO^+^CCR7^−^) were increased in the presence of treatment with platelet-derived mitochondria + SDF-1 Ab (Figure 6C). Thus, these data confirmed the interaction of SDF-1 and CXCR4 contributed the immune modulation of platelet-derived mitochondria on CD4^+^ T cells.

### 2.4. Phenotypic Comparison between Platelet-Derived Mitochondria with Other Cell-Derived Mitochondria

To determine whether the immune modulation of platelet-derived mitochondria was unique, we compared the differences among platelet-derived mitochondria (Plt-Mito), human peripheral blood mononuclear cells (PBMC)-derived mitochondria (PBMC-Mito), and human peripheral blood insulin-producing cells-derived mitochondria (PB-IPC-Mito) [21,22]. In comparison with platelet-derived mitochondria, PBMC-derived mitochondria displayed the similar levels of SDF-1 expression as that of platelet-derived mitochondria, with no significant differences (Figure 7A,B, 77.4% ± 5.12% vs. 73.7% ± 2.0%, *p* = 0.47). However, the level of SDF-1 expression on the PB-IPC-Mito was markedly lower than that of platelet-derived mitochondria (Figure 7A,B, *p* = 0.004). Additionally, we tested the modulation of PBMC-derived mitochondria on memory CD4^+^ T cells, the anti-CD3/CD28-activated CD4^+^ T cells were treated with PBMC-derived mitochondria for 48 h. Mitochondrion-untreated anti-CD3/CD28-activated CD4^+^ T cells served as control group. Flow cytometry revealed that the percentages of naïve CD4^+^ T cells (CD4^+^CD45RO^−^CCR7^+^) (Figure 7C) and T_CM_ cells (CD4^+^CD45RO^+^CCR7^+^) were markedly increased (Figure 7D), but the percentage of T_EM_ cells (CD4^+^CD45RO^+^CCR7^−^) was significantly decreased (Figure 7E) post treatment with PBMC-derived mitochondria. It indicates that PBMC-derived mitochondria display the similar effects of immune modulation as platelet-derived mitochondria.

### 2.5. Gene Expression Profiling of CD4^+^ T Cells after the Treatment with Platelet-Derived Mitochondria

To examine the gene expression profiling of CD4^+^ T cells, we performed the RNA sequencing (RNA-seq) analysis of anti-CD3/CD28-activated CD4^+^ T cells post treatment with platelet-derived mitochondria. In comparison with the control group, the data demonstrated that 53 genes were markedly down-regulated in the mitochondrion-treated group (Figure 8A,B, *p* < 0.05), while 25 genes were up-regulated (Figure 8A,C, *p* < 0.05). There were no significant changes for other genes (*n* = 17,102, 99.54% of genes) in anti-CD3/CD28-activated CD4^+^ T cells after the treatment with platelet-derived mitochondria.

Carboxypeptidase M (CPM) was highly expressed on the membrane of activated CD4^+^ T cells [23], contributing to the cleavage of C-terminal arginine (Arg) or lysine (Lys) of growth factors or cytokine for releasing [24]. To confirm the function-associated gene expression from above RNA-seq data, we have analyzed the expression of carboxypeptidase M at gene and protein levels by using real time PCR and flow cytometry, respectively, in anti-CD3/CD28 Dynabead-activated CD4^+^ T cells after the treatment with platelet-derived mitochondria (Figure 8 D,E). The data demonstrated that both gene and protein expressions of CPM were markedly down-regulated in the presence of platelet-derived mitochondria.

## 3. Discussion

CD4^+^ T cells are the predominant cell population to orchestrate human innate and adaptive immune responses against infections, cancer formations, inflammations, and the developments of autoimmune diseases. Increasing clinical evidence demonstrated that dysfunctions of CD4^+^ T cells contributed to multiple chronic diseases such as type 1 diabetes and autoimmune diseases. The current study demonstrated that platelet-derived mitochondria act as novel immune modulators on the anti-CD3/CD28 bead-activated CD4^+^ T cells, which exhibited multiple changes including cell surface markers, functionality, cytokine productions, and gene expressions after mitochondrial treatment. The data established the direct immune modulation of platelet-derived mitochondria on CD4^+^ T cells through the SDF-1 and its ligand CXCR4.

Based on the co-localization of MitoTracker Deep Red-positive platelet-derived mitochondria and MitoTracker Green-positive intrinsic mitochondria in CD4^+^ T cells, together with the significant up-regulation of mtDNA-specific gene markers ND1 and ND5 in CD4^+^ T cells, these data suggest that platelet-derived mitochondria may enter into CD4^+^ T cells. This was similar to our previous work on the penetration of platelet-derived mitochondria into human peripheral blood-derived insulin-producing cells (PB-IPC) [21]. There is a high expression of CXCR4 on CD4^+^ T cells, contributing to the migration of CD4^+^ T cells to pancreatic islets and destruction of β cells in T1D [25,26] and other autoimmune diseases [27]. Due to the expression of CXCR12 (SDF-1) on platelet-derived mitochondria [21], they may target these pathogenic CD4^+^ T cells through the interaction of CXCR4/CXCRL12. Importantly, platelet-derived mitochondria display the immune tolerance-associated markers including programmed death ligand-1 (PD-L1) and CD270 [16], while CD4^+^ T cells present their ligands PD-1 and B- and T- lymphocyte attenuator (BTLA), respectively, mitochondria may modulate CD4^+^ T cells through the interactions of PD-1/PD-L1 and BTLA/CD270, respectively. This highlights the translational potentials of platelet-derived mitochondria for the treatment of autoimmune diseases in clinics. Since the platelet numbers were increased in T1D subjects after receiving SCE therapy [16], which may release more mitochondria into blood circulation [17], the immunomodulation of platelet-derived mitochondria provides additional mechanisms underlying the SCE therapy for the treatment of T1D and other autoimmune diseases.

Based on different surface markers (e.g., CCR7 and CD45RA or CD45RO), CD4^+^ T cells are sub-divided into Naïve T cells, T_CM_, T_EM_, and regulatory T cells (Tregs) [5]. Naïve T cells are the pool of precursors that have the capability to give rise to effector and memory T cells upon receiving the antigen signals from antigen-presenting cells (APC). Spainer et al. reported the increased insulin-specific CD4^+^ T_EM_ cells in recent onset T1D patients [2]. Matteucci et al. reported that both percentages and absolute numbers of naïve and T_CM_ cells were reduced, while the terminally differentiated effector memory T cells were markedly increased in T1D patients comparing with the healthy control [28]. The persistence of autoreactive memory T cells leads to the difficulty in rescuing the residual islet β cells in new-onset T1D and improving the therapeutic potentials of islet transplants [4]. Therefore, it will be essential to eliminate these autoimmune memory T cells and avoid the persistent destructions of the transplanted islet β cells and/or newly-generated insulin-producing cells. The current study demonstrated the up-regulated percentages of CD4^+^ Naïve T and CD4^+^ T_CM_ cells, and the reduced percentage of CD4^+^ T_EM_ cells in the anti-CD3/CD28 bead-activated CD4^+^ T cells after the treatment with platelet-derived mitochondria. Current ex vivo data were similar to our previous clinical studies in T1D patients after receiving Stem Cell Educator therapy [14]. Additional data demonstrated the direct immune modulation of platelet-derived mitochondria in the anti-CD3/CD28 bead-activated CD4^+^ T cells such as down-regulation of inflammation-associated cytokines (e.g., IL-6 and IL-21), up-regulation of Th2-associated cytokine IL-4 but down-regulation of IL-5, and changes in gene profiling. For instance, carboxypeptidase M (CPM) was highly expressed on the membrane of activated CD4^+^ T cells [23], contributing to the cleavage of C-terminal arginine (Arg) or lysine (Lys) of growth factors or cytokine for releasing [24]. Importantly, both gene and protein expressions of CPM were markedly down-regulated in the presence of platelet-derived mitochondria, which were consistent with the suppression of CD4^+^ T cells after the treatment with platelet-derived mitochondria. The data provides new molecular mechanisms about mitochondrial immune modulation involved in the SCE therapy.

Similar to most autoimmune diseases, the autoimmunity of T1D is complicated and involves different compartments of the immune system including CD4^+^, CD8^+^ T cells, Tregs, B cells, DCs, monocyte/macrophages (Mo/Mϕs), and natural killer T cells (NKTs). Consequently, most efforts to develop successful treatments or a cure for T1D have been hindered over the last 40 years. Mitochondria circulate in human and animal peripheral blood [17]. Flow cytometry proved that they may directly interact with monocytes, DC, B cells and other cell compartments, except CD4^+^ T cells. As novel immune modulators, circulating mitochondria may play a central role in the induction of immune tolerance and balance. Additionally, platelets are the second largest cell population in human blood, without a nucleus. The highly-purified mitochondria derived from platelets offer a promising research tool to explore their immunomodulatory effects and translational potentials for the clinical treatment of human diseases.

## 4. Materials and Methods

### 4.1. PBMC and CD4^+^ T Cells Collection

Human buffy coat blood units (*n* = 26; mean age of 44 ± 16.8; age range from 16 to 66 years old; 17 males and 9 females) were purchased from the New York Blood Center (New York, NY, USA, http://nybloodcenter.org/). Human buffy coats were added to 40 mL of chemical-defined serum-free culture X-VIVO 15^TM^ medium (Lonza, Walkersville, MD, USA) and mixed with 10 mL pipette. Next, they were used for isolation of peripheral blood-derived mononuclear cells (PBMC). Mononuclear cells were isolated from buffy coats blood by Ficoll-Paque^TM^ PLUS (*γ* = 1.007, GE Healthcare, Chicago, IL, USA), followed by the removal of the red blood cells using Red Blood Cell Lysis buffer (eBioscience, San Diego, CA, USA). After three washes with saline, the whole PBMC were seeded in chemical-defined serum-free culture X-VIVO 15^TM^ medium (Lonza, Walkersville, MD, USA), without adding any other growth factors, and incubated at 37 °C in 8% CO_2_ conditions.

To get the purified CD4^+^ T cells, PBMC were stained with CD4-FITC antibody for 30 min and purified with Anti-FITC Magnetic Beads (Miltenyi Biotech, Gladbach, Germany) according to the manufacturer’s instructions.

### 4.2. Isolation of Mitochondria from Platelets and Other Cells

Human platelets samples (*n* = 15) were separated from adult volunteer donors and purchased at New York Blood Centers (New York, NY, USA, http://nybloodcenter.org/). The mitochondria were isolated from PB-platelets by a Mitochondria Isolation kit (Thermo scientific, Rockford, IL, USA, Prod: 89874) according to the manufacturer’s recommended protocol [16]. The protein concentration of mitochondria was measured by a NanoDrop 2000 Spectrophotometer (ThermoFisher Scientific, Waltham, MA, USA). The isolated mitochondria were aliquoted and kept in a −80 °C freezer.

Platelets-derived mitochondria were stained with MitoTracker Deep Red FM (100 nM) (Thermo Fisher Scientific, Waltham, MA, USA), at 37 °C for 15 min according to the manufacturer’s protocol, followed by three washes with PBS at 12,000 rpm × 15 min at 4 °C. CD4^+^ T cells were stained with MitoTracker Green FM (100 nM) (Thermo Fisher Scientific, Waltham, MA, USA) at 37 °C for 15 min according to the manufacturer’s protocol, followed by three washes with PBS at 300× *g* at 4 °C for 10 min.

To compare the phenotypic differences of platelet-derived mitochondria with other cells, mitochondria were isolated from PBMC and peripheral blood-derived insulin-producing cells (PB-IPC) by using the same Mitochondria Isolation kit (Thermo Fisher Scientific, Waltham, MA, USA) according to the manufacturer’s recommended protocol. Culture of PB-IPC from adult peripheral blood was performed as previously described [21,22]. Briefly, the whole PBMC were seeded in 150 × 15 mm Petri dishes (BD Falcon, NC, USA) at 1 × 10^6^ cells/mL, 25mL/dish in chemical-defined serum-free culture X-VIVO 15^TM^ medium (Lonza, Walkersville, MD, USA) without adding any other growth factors and incubated at 37 °C in 8% CO_2_ [29]. Seven days later, PB-IPC were growing and expanded by adhering to the hydrophobic bottom of Petri dishes for experiments.

### 4.3. PBMC Proliferation Assay

PBMC were labeled with carboxyfluorescein succinimidyl ester (CFSE) (Thermo Fisher Scientific, Waltham, MA, USA) according to the manufacturer’s instructions [30]. Next, they were cocultured with Dynabeads coupled with anti-CD3 and anti-CD28 antibodies (Thermo Fisher Scientific, Waltham, MA, USA) for 72 h in the presence of mitochondria at 25 μg/mL, 50 μg/mL, 100 μg/mL and 200 μg/mL, respectively. Lastly, they were incubated at 37 °C in 5% CO_2_. Mitochondria-untreated cells served as control.

### 4.4. Flow Cytometry

For surface staining, samples were pre-incubated with human BD Fc Block (BD Pharmingen, Franklin Lakes, NJ, USA) for 15 min at room temperature and then directly aliquoted for different antibody staining [16]. Cells were stained with antibody for 30 min at room temperature and then washed with PBS at 300× *g* for 10 min before flow analysis. Cells were next incubated with different mouse anti-human mAb from Beckman Coulter (Brea, CA, USA), including PE-conjugated anti-CD56 and anti-CCR7, PE-Cy5-conjugated anti-CD19, PE-Cy7-conjugated anti-CD11c and anti-CD45RO, APC-Alexa Fluor 750-conjugated anti-CD8, and Krome Orange-conjugated anti-CD14. Anti-human mAb from BD Biosciences (BD Biosciences, San Jose, CA, USA) included the following: APC-conjugated anti-human CD4 antibody and propidium iodide (PI). Pacific Blue (PB)-conjugated anti-human CD3 Ab, FITC-conjugated anti-human Hsp60, and AlexaFluor-488-conjugated anti-human Cytochrome C were from BioLegend (San Diego, CA, USA). Anti-SDF-1 polyclonal mAB was purchased from Abcam (Cambridge, MA, USA). Anti-carboxypeptidase M (CPM) monoclonal mAB was purchased from Novocastra (Newcastle, UK). The FITC-conjugated AffiniPure donkey anti-mouse 2nd Ab was from Jackson ImmunoResearch Laboratories (West Grove, PA, USA). AlexaFluor-546-conjugated Calnexin and isotype-matched IgG and AlexaFluor-647-conjugated GM130 were purchased from Santa Cruz Biotechnology (Dallas, TX, USA). For intra-cellular and intra-mitochondrial staining, cells or mitochondria were fixed and permeabilized according to the PerFix-nc kit (Beckman Coulter, Brea, CA, USA) manufacturer’s recommended protocol. Isotype-matched mouse anti-human IgG antibodies (Beckman Coulter, Brea, CA, USA) served as a negative control. After staining, cells were collected and analyzed by a Gallios Flow Cytometer (Beckman Coulter, Brea, CA, USA) equipped with three lasers (488 nm blue, 638 red, and 405 violet lasers) for the concurrent reading of up to 10 colors. The final data were analyzed using the Kaluza Flow Cytometry Analysis Software version 2.1 (Beckman Coulter, Brea, CA, USA).

To determine the direct modulation of mitochondria on CD4^+^ T cells, the purified CD4^+^ T cells (1 × 10^5^ cells/well) from PBMC of healthy donors (*n* = 6) were activated with T-cell activator anti-CD3/CD28 Dynabeads in the presence or absence of 100 μg/mL platelet-derived mitochondria (*n* = 3) or PBMC-derived mitochondria (*n* = 3), respectively. Untreated CD4^+^ T cells served as negative control. Cells were collected for flow cytometry analysis after the treatment at 37 °C and 5% CO_2_ for 48 h.

### 4.5. Confocal Detection for Interaction between Mitochondria and CD4^+^ T Cells

To explore the interaction of mitochondria and CD4^+^ T cells, the platelets derived mitochondria were stained with MitoTracker Deep Red FM (100 nM) (Thermo Fisher Scientific, Waltham, MA, USA) and cocultured with MitoTracker Green FM (100 nM) (Thermo Fisher Scientific, Waltham, MA, USA) labeled CD4^+^ T cells for 2 h, at room temperature. Hoechst 33,342 (Sigma, Saint Louis, MO, USA) was used to stain the nuclei. Briefly, the purified CD4^+^ T cells were initially labeled with MitoTracker Green FM (100 nM) (Thermo Fisher Scientific, Waltham, MA, USA) and Hoechst 33,342 (Sigma, Saint Louis, MO, USA), and then planted in the 96-well tissue culture-treated plates at 2 × 10^5^ cells/well with serum-free culture medium X-VIVO 15 in the presence or absence of MitoTracker Deep Red-labeled platelet-derived mitochondria. The interaction between mitochondria and CD4^+^ T cells was directly observed and photographed under a confocal microscope with Nikon A1R confocal microscope on Nikon Eclipse Ti2 inverted base at room temperature.

### 4.6. Quantitative Real Time PCR

Total RNAs from each sample were extracted using a Qiagen kit (Valencia, CA, USA). First-strand cDNAs were synthesized from total RNA using an iScript gDNA Clear cDNA Synthesis Kit according to the manufacturer’s protocol (Bio-Rad, Hercules, CA, USA). The quantification of human mitochondrial DNA by real-time PCR was performed by using the StepOnePlus Real-Time PCR System (Applied Biosystem, CA, USA) [16]. Human mitochondrial DNA (mtDNA) monitoring primer sets (catalogue No. 7246), including ND1 and ND5 (Takara Bio, Mountain view, CA, USA), were used to detect the mitochondrial DNA. SLCO2B1 Primer Mix and SERPINA1 Primer Mix (Takara Bio, Mountain view, CA, USA) were used for detection of nuclear DNA (nDNA). The data analysis of the mtDNA Copy Number was performed by using the mtDNA Copy Number Calculation Tool (https://www.takarabio.com/resourcedocument/x102669), as recommended by the manufacturer.

The validated primers of Carboxypeptidase M (CPM) (Catalogue No. PPH17190E-200) and β-actin (Catalogue No. PPH00073G-200) were purchased from Qiagen (Valencia, CA, USA) and running under following conditions: 95 °C for 10 min, then 40 cycles of 95 °C for 15 s, and 60 °C for 60 s. The expression level of Carboxypeptidase M was determined relative to β-actin as an internal control.

### 4.7. Blocking Experiments with CXCR4 Receptor Antagonist AMD 3100 and SDF-1 Antibody

To determine whether the SDF-1/CXCR4 signal was contributing to the interaction between mitochondria and CD4^+^ T cells, we performed the blocking experiments with CXCR4 receptor antagonist AMD3100 and SDF-1 Ab. The isolated CD4^+^ T cells were treated with MitoTracker Deep Red-labeled purified mitochondria in the presence or absence of AMD 3100 (30 μM) for 24 h. The equal concentration of DMSO served as control. The isolated CD4^+^ T cells were treated with MitoTracker Deep Red-labeled purified mitochondria in the presence or absence of SDF-1 Ab (20 μg/mL) for 2 h. To determine the blocking effects of SDF-1 Ab on the immune modulation of platelet-derived mitochondria, the purified CD4^+^ T cells (1 × 10^5^ cells/well) from PBMC of healthy donors (*n* = 3) were activated with T-cell activator anti-CD3/CD28 Dynabeads in the presence or absence of 100 μg/mL platelet-derived mitochondria or 20 μg/mL SDF-1 Ab. Untreated CD4^+^ T cells served as negative control. Cells were collected for flow cytometry analysis after the treatment at 37 °C and 5% CO_2_ for 48 h. CD4^+^ T cells were washed twice with PBS and prepared for flow cytometry detection.

### 4.8. Evaluation of Cytokine Levels in Serum

To detect the cytokine production by CD4^+^ T cells, 1 × 10^5^ CD4^+^ T cells were stimulated with anti-CD3/anti-CD28 beads (Thermo Fisher Scientific, Waltham, MA, USA) in the presence or absence of mitochondria at 100 μg/mL in a 96-well plate with a total of 200 μL X-VIVO 15 serum-free culture medium (Lonza, Walkersville, MD, USA) per well. After the treatment for 48 h, the supernatants were collected to examine the level of inflammatory cytokines (IL-2, IL-4, IL-5, IL-6, IL-9, IL-10, IL-13, IL-17F, IL-21, IL-22, IFN-γ, and TNF-α) using a LEGENDplex™ Human Th Panel (Biolegend, San Diego, CA, USA) by Gallios Flow Cytometer according to the manufacturer’s protocols.

### 4.9. Western Blot

Purified mitochondria or cells were treated with RIPA buffer, as previously described [16,30]. Proteins (20 μg/sample) were separated by 4–15% Criterion Tris-HCl gel (Bio-Rad, Hercules, CA, USA) and transferred to the PVDF membrane, blotted overnight with mouse anti-human cytochrome C (Biolegend, Dedham, MA, USA) monoclonal antibody (mAb) and mouse anti-human Calnexin mAb (Santa Cruz, Dallas, TX, USA), and followed by anti-mouse HRP-conjugated secondary mAb (Cell Signaling, Danvers, MA, USA). Membranes were incubated with chemiluminescent substrate (Thermo Scientific, Rockford, IL, USA) and chemiluminescent signal was detected by using ChemiDoc Imaging System (Bio-Rad, Hercules, CA, USA). β-actin served as an internal loading control [31].

### 4.10. RNA-seq

RNA sequencing (RNA-seq) analysis was performed between the mitochondrion-treated and untreated CD4^+^ T cells in the presence of anti-CD3/anti-CD28 beads stimulation in three preparations, as previously described [21]. Total RNAs from each sample were extracted by a Qiagen kit (Valencia, CA, USA) and shipped to Genewiz (South Plainfield, NJ, USA) in dry ice for standard RNA sequencing and profiling gene expression by using Illumina NovaSeq™ 6000 Sequencing System (Genewiz, South Plainfield, NJ, USA), with 2 × 150 bp configuration, single index, per lane.

### 4.11. Statistical Analysis

Statistical analyses of data were performed with GraphPad Prism 8 (version 8.0.1) software. The normality test of samples was evaluated using the Shapiro–Wilk test. Statistical analyses of data were performed using the two-tailed paired Student’s t-test to determine statistical significance between untreated and treated groups. The Mann–Whitney U test was utilized for non-parametric data. Values were given as mean ± SD (standard deviation). Statistical significance was defined as *p* < 0.05. For RNA-seq data analysis, the Wald test with DESeq2 software was utilized to generate *p*-values and log2 fold changes. Genes with a *p*-value < 0.05 and absolute log2 fold change >1 were determined as differentially expressed genes for each comparison.

## Figures and Tables

**Figure 1 ijms-21-06295-f001:**
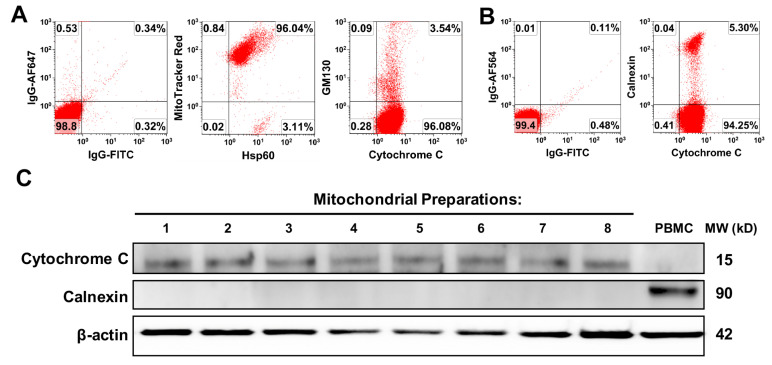
The purity and quality analysis of purified platelet-derived mitochondria. (**A**,**B**) The different organelle-specific markers have been utilized by flow cytometry such as MitoTracker Deep Red staining, anti-cytochrome C and anti-heat shock protein (HSP) 60 Abs for mitochondria, calnexin for endoplasmic reticulum (ER), and GM130 for Golgi apparatus. Isotype-matched IgGs served as negative controls (*n* = 3). (**C**) Western blotting showed the expression of cytochrome C in the purified platelet-derived mitochondria (*n* = 8). Peripheral blood mononuclear cells (PBMC) lysate served as control.

**Figure 2 ijms-21-06295-f002:**
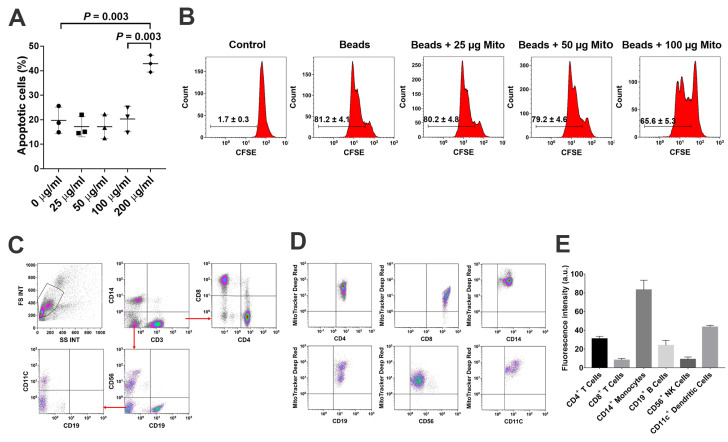
Suppression of PBMC proliferation by platelet-derived mitochondria (**A**) Apoptotic effects of PBMC after the treatment with different dosages of platelet-derived mitochondria. (**B**) Suppression of PBMC proliferation by platelet-derived mitochondria. The carboxyfluorescein succinimidyl ester (CFSE)-labeled PBMC were stimulated to proliferate with T-cell activator anti-CD3/CD28 Dynabeads in the presence of different dosages of platelet-derived mitochondria. Untreated PBMC served as negative control. Histograms of flow cytometry were representative of three experiments with similar results. (**C**) Gating strategy for flow cytometry analysis with the lineage-specific surface markers for different cell populations in PBMC (*n* = 3), including CD3/CD4/CD8 for subsets of T cells, CD19 for B cells, CD14 for monocytes, CD11c for dendritic cells (DCs), and CD56 for NK cells. (**D**) Flow cytometry revealed the distributions of MitoTracker Deep Red-labeled mitochondria (*n* = 3) among different cell populations. (**E**) Different types of immune cells displayed different levels of median fluorescence intensity. The data were given as mean ± SD of three PBMC (*n* = 3) treated with two preparations of platelet-derived mitochondria (*n* = 2).

**Figure 3 ijms-21-06295-f003:**
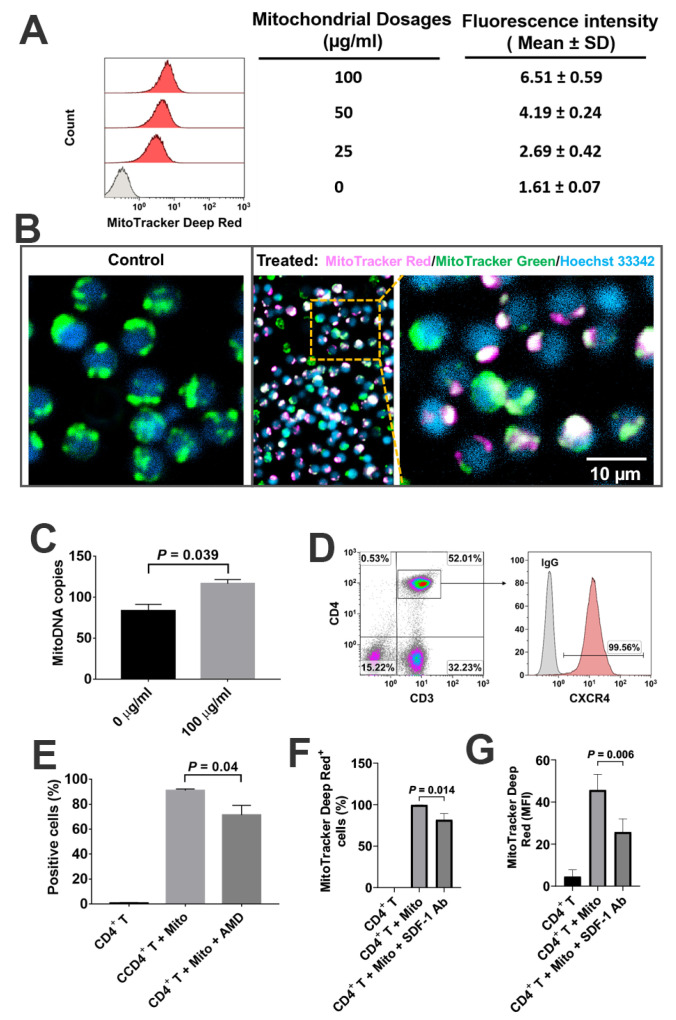
Interaction of platelet-derived mitochondria with CD4^+^ T cells. CD4^+^ T cells were purified from PBMC (*n* = 3) by using CD4^+^ microbeads (Miltenyi Biotec), with purity of CD4^+^ T cells > 95%. (**A**) Increased in the median fluorescence intensity of CD4^+^ T cells after the treatment with different dosages of MitoTracker Deep Red-labeled platelet-derived mitochondria. CD4^+^ T cells (1 × 10^5^ cells/well) were treated with different dosages of mitochondria for 24 h at 37 °C in 5% CO_2_ conditions, and then performed for flow cytometry. Data represent mean ± SD. *n* = 3. (**B**) Confocal microscopy showed the interaction of MitoTracker Deep Red-labeled mitochondria (pink) with CD4^+^ T cells (green color for intrinsic mitochondria of CD4^+^ T cells; blue, nuclear staining with Hoechst 33342), with a high magnification showing the distribution of platelet-derived mitochondria (pink) in the CD4^+^ T cells after the co-incubation for 2 h at room temperature. *n* = 3. (**C**) Numbers of mitoDNA copies were increased after the treatment of CD4^+^ T cells with 100 μg/mL platelet-derived mitochondria. Data represent mean ± SD. *n* = 3. (**D**) Expression of CXCR4 on gated CD3^+^ CD4^+^ T cells. Isotype-matched IgGs served as control. *n* = 3. (**E**) Flow cytometry indicated that the percentage of MitoTracker Deep Red-labeled mitochondria-positive CD4^+^ T cells were declined after blocking with 30 µM AMD3100 for 24 h. Data represent mean ± SD. *n* = 3. (**F**) Flow cytometry indicated that the percentage of MitoTracker Deep Red-labeled mitochondria-positive CD4^+^ T cells were reduced in the presence of treatment with 20 μg/mL stromal cell-derived factor-1 (SDF-1) Ab at 37 °C and 5% CO_2_ for 2 h. *n* = 3. (**G**) Flow cytometry indicated that the median fluorescence intensity (MFI) of MitoTracker Deep Red-labeled mitochondria-positive CD4^+^ T cells was markedly declined in the presence of treatment with 20 μg/mL SDF-1 Ab at 37 °C and 5% CO_2_ for 2 h. Data represent mean ± SD. *n* = 3.

**Figure 4 ijms-21-06295-f004:**
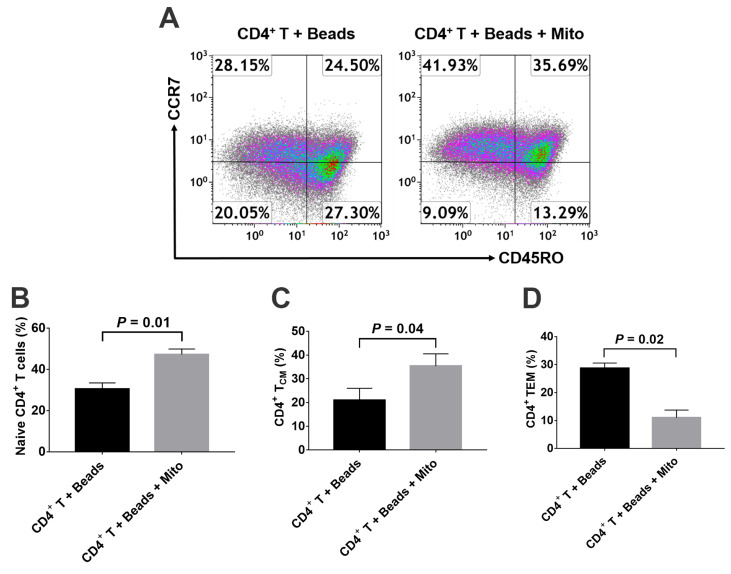
Effects of platelet-derived mitochondria on the differentiation of memory CD4^+^ T cells. The purified CD4^+^ T cells (1 × 10^5^ cells/well) from PBMC of healthy donors (*n* = 3) were activated with T-cell activator CD3/CD28 Dynabeads in the presence or absence of 100 μg/mL platelet-derived mitochondria (*n* = 3). Untreated CD4^+^ T cells served as negative control. (**A**) Flow cytometry analysis with human memory CD4 T cell-associated markers CD45RO and CCR7, with three subsets including CD45RO^−^CCR7^+^ Naïve T cells, CD45RO^+^CCR7^+^ T_CM_ cells, and CD45RO^+^CCR7^−^ T_EM_ cells. Dot plots of flow cytometry were representative of three experiments with similar results. (**B**) Upregulation of the percentage of naïve CD4^+^ T cells by platelet-derived mitochondria. (**C**) Increase in the percentage of CD4^+^ T_CM_cells by platelet-derived mitochondria. (**D**) Decrease in the percentage of CD4^+^ T_EM_ cells by platelet-derived mitochondria. Data represent mean ± SD. *n* = 3.

**Figure 5 ijms-21-06295-f005:**
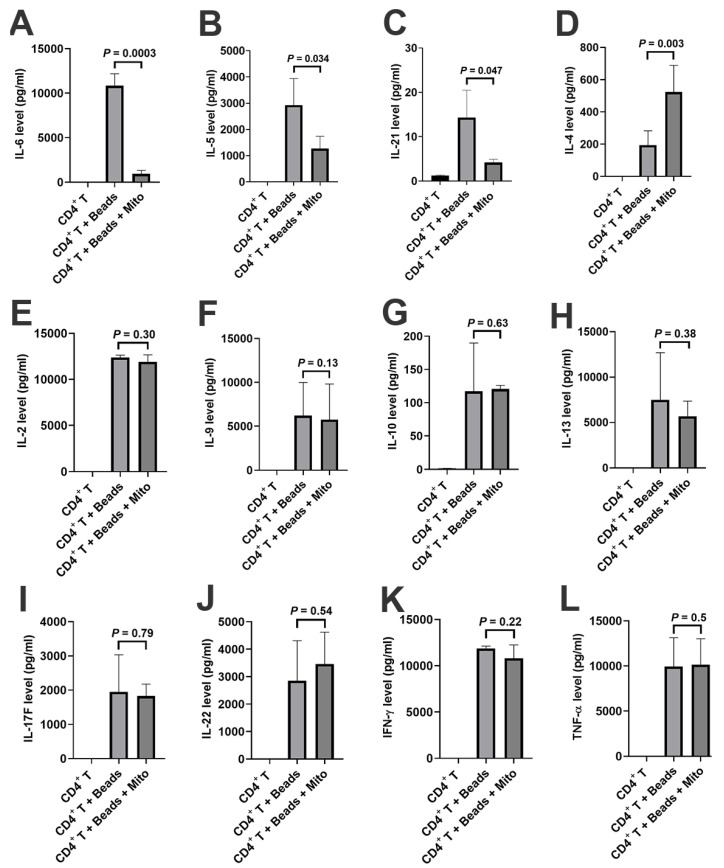
Modulation of cytokine productions in CD4^+^ T cells by platelet-derived mitochondria. The purified CD4^+^ T cells (1 × 10^5^ cells/well) from PBMC of healthy donors (*n* = 3) were activated with T-cell activator anti-CD3/CD28 Dynabeads in the presence or absence of 100 μg/mL platelet-derived mitochondria (*n* = 3). Untreated CD4^+^ T cells served as negative control. After the treatment for 2 days, supernatants were collected to measure the levels of cytokines (IL-2, IL-4, IL-5, IL-6, IL-9, IL-10, IL-13, IL-17F, IL-21, IL-22, IFN-γ, and TNF-α) by using a LEGENDplex™ Human Th Panel (Biolegend) with Gallios Flow Cytometer. (**A**) The level of IL-6 was markedly decreased after the mitochondrial treatment. (**B**) The level of IL-5 was markedly decreased after the mitochondrial treatment. (**C**) The level of IL-21 was significantly decreased after the mitochondrial treatment. (**D**) The level of IL-4 was significantly increased after the mitochondrial treatment. (**E**–**L**) There were no marked differences in the levels of IL-12 (**E**), IL-9 (**F**), IL-10 (**G**), IL-13 (**H**), IL-17F (**I**), IL-22 (**J**), IFN-γ (**K**), and TNF-α (**L**) after the treatment with platelet-derived mitochondria. Data were given as mean ± SD (*n* = 3).

**Figure 6 ijms-21-06295-f006:**
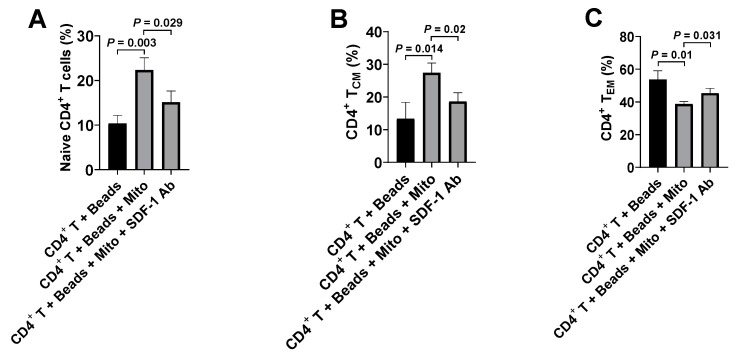
Blocking effects of SDF-1 Ab on the immune modulation of platelet-derived mitochondria. The purified CD4^+^ T cells (1 × 10^5^ cells/well) from PBMC of healthy donors (*n* = 3) were activated with T-cell activator CD3/CD28 Dynabeads in the presence or absence of 100 μg/mL platelet-derived mitochondria or 20 μg/mL SDF-1 Ab. Untreated CD4^+^ T cells served as negative control. Cells were collected for flow cytometry analysis after the treatment at 37 °C and 5% CO_2_ for 48 h. (**A**) The percentage of CD45RO^−^CCR7^+^ Naïve T cells were down-regulated in the presence of SDF-1 Ab. (**B**) The percentage of CD45RO^+^CCR7^+^ T_CM_ cells were decreased after blocking with SDF-1 Ab. (**C**) The percentage of CD45RO^+^CCR7^−^ T_EM_ cells were up-regulated in the presence of SDF-1 Ab. Data represent mean ± SD. *n* = 3.

**Figure 7 ijms-21-06295-f007:**
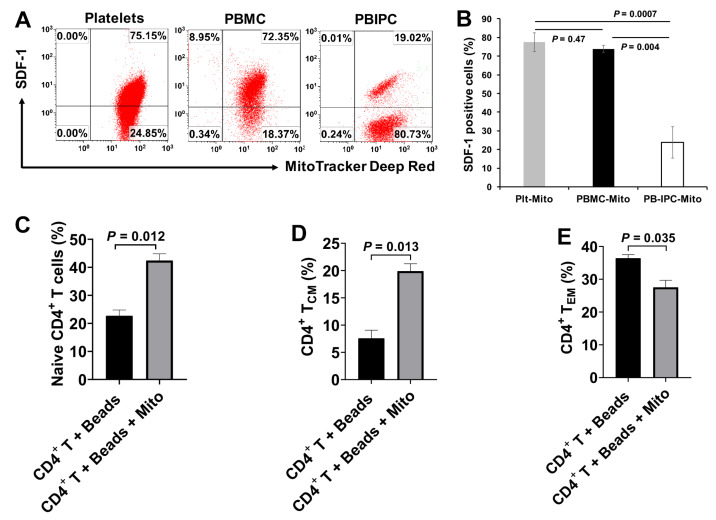
Compare platelet-derived mitochondria with other cell-derived mitochondria. (**A**,**B**) Compare the SDF-1 expression among platelet-derived mitochondria, PBMC-derived mitochondria and human peripheral blood insulin-producing cells (PB-IPC)-derived mitochondria. The mitochondria were isolated from peripheral blood (PB)-platelets, PBMC and PB-IPC, respectively, using the Mitochondria Isolation kit (Thermo scientific) according to the manufacturer’s recommended protocol. Culture of PB-IPC from adult peripheral blood was performed as previously described [21,22]. (**A**) Flow cytometry show the expression of SDF-1 on the platelet-derived mitochondria and PBMC-derived mitochondria and PB-IPC-derived mitochondria. Isotype-matched IgGs served as controls. Data were representative from three experiments. (**B**) Compare the level of SDF-1 expression between platelet-derived mitochondria (Plt-Mito) and PBMC-derived mitochondria (PBMC-Mito) with no marked difference, but much lower on PB-IPC-Mito. Data represent mean ± SD, *n* = 3. (**C**–**E**) Effects of PBMC-derived mitochondria on the differentiation of memory CD4^+^ T cells. The purified CD4^+^ T cells (1 × 10^5^ cells/well) from PBMC of healthy donors (*n* = 3) were activated with T-cell activator anti-CD3/CD28 Dynabeads in the presence or absence of 100 µg/mL PBMC-derived mitochondria (*n* = 3). Untreated CD4^+^ T cells served as negative control. (**C**) Upregulation of the percentage of naïve CD4^+^ T cells by PBMC-derived mitochondria. (**D**) Increase in the percentage of CD4^+^ T_CM_ cells by PBMC-derived mitochondria. (**E**) Decrease in the percentage of CD4^+^ T_EM_ cells by PBMC-derived mitochondria. Data represent mean ± SD. *n* = 3.

**Figure 8 ijms-21-06295-f008:**
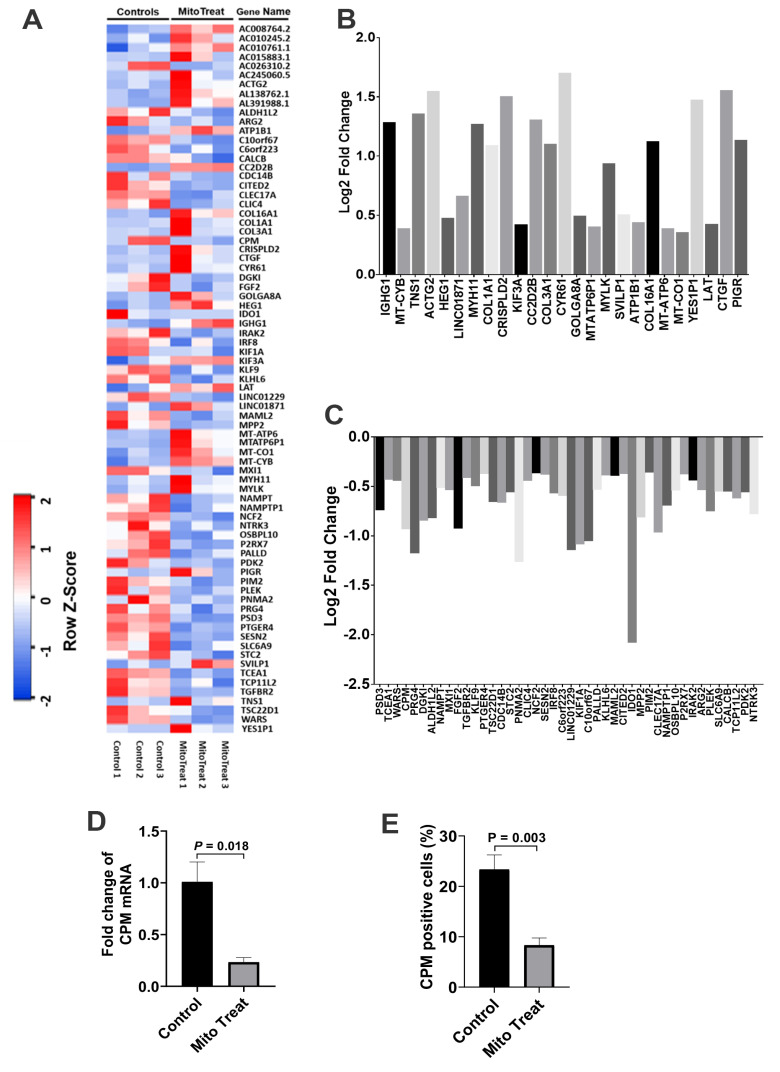
RNA-seq for gene expression profiling of CD4^+^ T cells after the treatment with platelet-derived mitochondria. (**A**) The heatmap showed differentially expressed genes shown by the RNA-seq in three anti-CD3/CD28 bead-activated CD4^+^ T cells in the absence (control, left three columns) and presence of platelet-derived mitochondria (mitoTreat, right three columns). (**B**) The heatmap revealed 25 up-regulated genes in CD4^+^ T cells post the treatment with platelet-derived mitochondria. (**C**) Fifty-three down-regulated genes in CD4^+^ T cells post the treatment with platelet-derived mitochondria. (**D**,**E**) The purified CD4^+^ T cells (1 × 10^6^ cells/mL in 6-well tissue culture plates) from PBMC of healthy donors (*n* = 3) were activated with T-cell activator anti-CD3/CD28 Dynabeads in the presence or absence of 100 μg/mL platelet-derived mitochondria. Untreated CD4^+^ T cells served as negative control. Cells were collected for testing after the treatment at 37 °C and 5% CO_2_ for 48 h. (**D**) Down-regulation of Carboxypeptidase M (CPM) mRNA level, as indicated by real time PCR. (**E**) Flow cytometry demonstrated the decrease in the percentage of CPM-positive cells. Data represent mean ± SD. *n* = 3.

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
