# Peer review of "Immune Modulation of Platelet-Derived Mitochondria on Memory CD4+ T Cells in Humans"

_ijms, 2020, doi:10.3390/ijms21176295_

Round 1
Reviewer 1 Report
The manuscript by Yu et al. reports changes in phenotype and gene expression of human CD4 T cells treated with platelet-derived mitochondria and suggest that this might be a useful therapy for autoimmune diseases such as T1D.
Overall, I found the scientific question investigated interesting. The studies are acceptable and appropriate controls are included. The analysis of the presented data is appropriate; however, the interpretation of the data has needs further clarification.
- The manuscript does not provide a possible mechanism which explains how the mito are affecting T cells (the authors suggest that it is a T cell-intrinsic mechanism following mito integration into the cell). This should be discussed either in the intro or in the discussion and put in context of the findings provided in this manuscript.
- In figure 2 the authors suggested that the changes observed in T cells are due to binding of mito to T cell via CXCR4/CXCL12. How does this integration then lead to modulation of the mito-obtaining cell? Moreover, the author suggested that this integration is CXCR4 dependent, however, they fail to provide compelling evidence for that. Can the author use blocking Abs for CXCR4 and/or CXCL12? The chemical inhibitor (such as AMD3100) may have off target effects and can be toxic to the cells, and since there are antibodies available to target human cells, I think this is more appropriate. Additionally, the author did not provide evidence that CXCL12 protein is expressed on the surface of the isolated mitochondria. Can the author also provide an image as in 2B showing mito+cells after CXCR4/CXCL12 blockade? Furthermore, is it also possible that the integration is through trogocytosis? Moreover, if the mito are expressing CXCL12, which then bind to CXCR4 on the surface of T cells, wouldn’t that then lead to CXCR4-mediated signaling? Can the author comment on that?
- Is it possible that the MitoTracker dye is being released/defused through the media rather than the cells integrating the mitochondria? This will then explain many of the observations in Figure 1. Along this line, can the author provide evidence that the mitochondria are still intact during the in vitro cultures?
- Are the mito being isolated from the same patient which the CD4 cells are taken from? i.e. are these “auto”-culture or “allo”-cultures?
- In figure 3 the authors show changes in cytokine expression following mito + CD4 T cell cultures. In Figure 4 they show changes in gene expression. I found it strange that none of the cytokines which are significantly altered in Figure 3 (e.g. IL-6) are shown in figure 4. Can the author explain this? It is interesting that there are no changes in IL-2 levels (Figure 3) although the mito + cells show reduced proliferation (Figure 1). Generally, cells that proliferate less also produce less cytokines, which fits with their observations for IL-6, IL-5, and IL-21 (all of which can also promote T cell proliferation) but not with IL-4 levels. Can the authors provide an explanation for these? Do the cells differentiation into a different T helper cell subset?
- Figure 4 show changes in gene expression of T cells cultured with mito. This figure is not being well connected to the overall story, but is rather seemed as an after-thought, and should be better integrated into the manuscript. What was the expression cut-off of gene showed in figure 4? a more robust differential gene expression analysis such as deseq2/edgeR may have been more appropriate. Additionally, are there any genes, or pathways, which can be linked to the changes shown in figures 1-3? and may explain reduced proliferation, T cell differentiation, and effector function?
- Mitochondria are known to be a reservoir of many inflammatory mediators, such as ATP and heat-shock proteins. Extracellular stimulation of T cells (and other immune cells) with these molecules will signals and modulate their function. This has been shown numerously in the literature. Can the authors comment on that? Why do the authors think that this IS NOT the mechanism by which the cells are responding to the mitochondria in the cultures? Isn’t it more possible that the mito rupture in the culture, thus releasing ATP, HSP, etc. which then signal to T cells?
- The authors need to provide more reasoning as to why their proposed mechanism can serve to treat T1D/autoimmune diseases, and put in in the context of the literature and their finding. Since their proposed mechanism is CXCR4/CXCL12 dependent, then the CXCR4/CXCL12 axis would have been protective in T1D, however the literature suggests otherwise (see for example PMID: 26300887, PMID: 17374136, and PMID: 26214690), can the authors comment on that?
Author Response
Dear Reviewer,
We sincerely appreciate your kind consideration and providing very constructive comments for us to improve the quality of this manuscript. We have performed additional experiments and provided responses to your comments point-by-point.
Best regards,
Yong

Reviewer 2 Report
This manuscript by Yu et al. concerns the investigation of immune modulating effects from platelet-derived mitochondria on CD4 T cells. The authors find that treatment of CD4 T cells with mitochondria isolated from blood plasma (and presumed to be from platelets) reduce effector memory T cells and increase naïve and central memory T cells. The authors attempt to identify the mechanism by which this modulation occurs. The work will be of interest to the immune modulation field. However, this manuscript has numerous major and minor deficiencies in design, presentation and interpretation which should be addressed before it is ready for publication.
Major concerns
- Even though the authors followed their previously published protocol, they have provided insufficient evidence of mitochondrial isolation in this study, which forms the basis of this entire paper. Off-target staining with fluorescent stains is a major concern. The authors should provide biochemical validation of the isolation (e.g., western blot staining of Cytochrome C).
- The authors emphasize repeatedly throughout the manuscript that the effects seen are due to “platelet-derived mitochondria” with the implication that these are unique effects of mitochondria from platelets, but not other cells. Can the authors demonstrate that platelet-derived mitochondria are unique in their immune modulating properties, in comparison with mitochondria isolated from other cells (which might, for example, be released during programmed cell death processes)?
- Fig 2B – the authors do not indicate the dose of mitochondria used for this figure, and should provide representative images of all 3 doses (25, 50, 100 µg/mL).
- Fig 2B – to distinguish off-target/leached staining of CD4 cell mitochondria, the authors should consider co-staining MitoTracker Deep Red-stained mitochondria with MitoTracker Green or Red and presenting the co-stained images so readers can appreciate the cell-intrinsic mitochondria morphology along with the mitochondria used to treat the cells. Untreated cells are also not exposed to MitoTracker Deep Red, so it is impossible to determine that the mitochondria visualized are indeed exogenous.
- Fig 2C – the authors indicate that both ND1 and ND5 are used to quantify copies of mtDNA but do not indicate which of the genes was used for the data in the figure.
- Fig 2C – how was the qPCR for mtDNA normalized?
- Fig 3 – the authors should demonstrate that this effect is reduced or blocked by preventing the interaction of the mitochondria with their isolated cells (e.g., by AMD3100 treatment as in Fig 2E).
- Specific primer sequences should be reported in the methods.
- No methods are provided to evaluate the statistical analysis on the RNA-seq data.
- RT-qPCR validation should be performed on select, statistically significant genes identified by the RNA-seq data.
- RNA-seq datasets should preferably be published into a public depository.
- Fig 5A – what happened in MitoTreat 1? Why is it so different from the rest of the MitoTreat group and does this apparent outlier treatment affect the statistical determination of the up- and downregulated genes?
Other
- The authors should acknowledge the contributions of Dache et al. and their role in the discovery of cell-free, respiratory-competent mitochondria in blood (https://doi.org/10.1096/fj.201901917RR) (see line 58).
- This may be an effect of language differences, but the term “resident” as in “resident memory cells” is far more common than “residential” as in “residential memory cells.” This applies throughout the manuscript.
- The phrasing used in lines 41-48 is unusual, particularly the specific section “with this patented technology.” This makes the manuscript read more like a marketing document than a primary data article. Rephrasing to reduce apparent bias is encouraged. It is appropriate to simply list the patent number in the Conflict of Interest section and remove any marketing-type language from the manuscript.
- List the specific clinical trial identifiers where possible (lines 45-48).
- It is confusing with the introduction and discussion focus so heavily on T1D when the actual study does not seem to have a particular focus on T1D, but rather looks at general effects on CD4 cells with platelet-derived mitochondria. None of the primary blood samples come from T1D patients, or if they are it is not indicated. The authors should expand their intro/discussion to include the relevance of their data to other immunopathologies in which CD4 T cells have been implicated (eg. viral infections, MS, EAE).
- The word “improved” is used where “increased” is more appropriate. “Improved” implies a beneficial effect, whereas the authors are simply measuring increases in, for example, staining or cytokine levels. This should be addressed (e.g., line 114 and 167).
- Line 240 – the term “wonderful research tool” is unusual and, again, reads like a marketing statement.
- More discussion is necessary around the observations made in Figure 4 and 5.
- Line 294 – PB-IPCs don’t appear to be a part of this paper and may be a carryover from another paper from this group.
Author Response

(The authors gave the same response as above.)

Round 2
Reviewer 2 Report
This is a revised manuscript by Yu et al concerning immune modulating effects of platelet-derived mitochondria on CD4 T cells. While the additional experimentation performed by the authors does not directly address my initial questions (e.g. the specific concern in characterizing isolated mitochondria by flow cytometry was addressed by, again, characterizing isolated mitochondria by flow cytometry), the authors do make an effort to improve their manuscript. That being said, I have some continued concerns:
- The supplemental files were not made available for review. I am not aware of the number of supplemental figures included, and only see supplemental figures S4, S5 and S6.
- The methods for Figure S4 are absent. This should be added, and confirmed for the rest of the supplemental figures.
- Figure 2 (new). The MitoTracker Green stain is very poor and not sufficient to make an assessment of the intrinsic T cell mitochondria versus the exogenously added mitochondria. In fact, in the merged panel for the treated row, it appears that 100% of the MitoTracker Green stain is colocalizing with the exogenous mitochondria. Unless the authors are claiming that the exogenous mitochondria are replacing the intrinsic mitochondria, then this figure is not representative of the presented story. This experiment should be repeated to fix this problem.
- The link to the mtDNA calculation tool should be included in the manuscript (https://www.takarabio.com/resourcedocument/x102669)
- If sequences are not available for the primers, then the specific catalog numbers should be included in the methods.
- Description of the Wald test in DESeq2 is an improvement for the analysis of the RNAseq, but does not completely answer the question of how the data were analyzed. What was the specific release of the reference transcriptome used? How were counts generated? Using Bowtie2? The authors should provide all details regarding how the RNAseq went from raw sequences on the Illumina sequencer to final data, as which algorithms are used can significantly affect alignment/count/analysis results.
- There is no need to wait until publication to deposit the RNAseq into the public repository (https://www.ncbi.nlm.nih.gov/sra), and this information is usually included in submitted manuscripts with an embargo until publication. The authors are urged to do this.
- Section 4.2 – the authors note that they spun their cells at 12,000 rpm to wash the MitoTracker? There is no mention of a density gradient involved, so there is concern about shear stress on cells at this speed versus a more standard speed of 1200-1600 rpm (~300 g). Might this explain the staining results in Figure 2? Is this a typo? The authors should confirm that the cells were indeed spun at this high speed.
Author Response
Dear Reviewer,
We sincerely appreciate your kind considerations and providing very constructive comments for us to improve the quality of this manuscript. We have performed additional experiments and provided responses to your comments point-by-point.
Additionally, we are in the process of preparation to submit our RNAseq data to SRA, with a submission #SUB7990661.
Best regards,
Yours sincerely,
Yong Zhao M.D., Ph.D.
